# The SLOGERT Framework for Automated Log Knowledge Graph Construction [*]

Andreas Ekelhart[1][0000−0003−3682−1364], Fajar J. Ekaputra[2][0000−0003−4569−2496], and Elmar Kiesling[1][0000−0002−7856−2113]

[1] WU (Vienna University of Economics and Business), Welthandelsplatz 1, 1020 Vienna, Austria `first.last@ai.wu.ac.at`
[2] TU Wien (Vienna University of Technology), Favoritenstraße 9-11/194, 1040 Vienna, Austria `fajar.ekaputra@tuwien.ac.at`

**Abstract.** Log files are a vital source of information for keeping systems running and healthy. However, analyzing raw log data, i.e., textual records of system events, typically involves tedious searching for and inspecting clues, as well as tracing and correlating them across log sources. Existing log management solutions ease this process with efficient data collection, storage, and normalization mechanisms, but identifying and linking entities across log sources and enriching them with background knowledge is largely an unresolved challenge. To facilitate a knowledge-based approach to log analysis, this paper introduces SLOGERT, a flexible framework and workflow for automated construction of knowledge graphs from arbitrary raw log messages. At its core, it automatically identifies rich RDF graph modelling patterns to represent types of events and extracted parameters that appear in a log stream. We present the workflow, the developed vocabularies for log integration, and our prototypical implementation. To demonstrate the viability of this approach, we conduct a performance analysis and illustrate its application on a large public log dataset in the security domain.

**Keywords:** Knowledge graphs · Log analysis · Log vocabularies · Graph modelling patterns.

## 1 Introduction

Log analysis is a technique to deepen an understanding of an operational environment, pinpoint root causes, and identify behavioral patterns based on emitted event records. Nearly all software systems (operating systems, applications, network devices, etc.) produce their own time-sequenced log files to capture relevant events. These logs can be used, e.g., by system administrators, security analysts,

---

[*] This work was sponsored by the Austrian Science Fund (FWF) and netidee SCI-ENCE under grant P30437-N31 and the Austrian Research Promotion Agency FFG under grant 877389 (OBARIS). The authors thank the funders for their generous support.

and software developers to identify and diagnose problems and conduct investigations. Typical tasks include security monitoring and forensics [38,23], anomaly detection [16,11,28], compliance auditing [39,25], and error diagnosis [44,6]. Log analysis is also a common issue more generally in other domains such as power systems security [35], predictive maintenance [40], workflow mining [3,4], and business/web intelligence [17,30].

To address these varied applications, numerous log management solutions have been developed that assist in the process of storing, indexing, and searching log data. However, investigations across multiple heterogeneous log sources with unknown content and message structures is a challenging and time-consuming task [41,22]. It typically involves a combination of manual inspection and regular expressions to locate specific messages or patterns [34].

The need for a paradigm shift towards a more structured approach and uniform log representations has been highlighted in the literature for a long time [21,19,34], but although various standardization initiatives for event representation were launched (e.g., [18,29,9,8]), none of them has seen widespread adoption. As a result, log analysis requires the interpretation of many different types of events, expressed with different terminologies, and represented in a multitude of formats [29], particularly in large-scale systems composed of heterogeneous components. As a consequence, the analyst has to manually investigate and connect this information, which is time consuming, error prone and potentially leads to an incomplete picture.

In this paper, we tackle these challenges and propose Semantic LOG ExtRaction Templating (SLOGERT), a framework for automated Knowledge Graph (KG) construction from unstructured, heterogeneous, and (potentially) fragmented log sources, building on and extending initial ideas [12]. The resulting KGs enable analysts to navigate and query an integrated, enriched view of the events and thereby facilitates a novel approach for log analysis. This opens up a wealth of new (log-structured) data sources for KG building.

Our main contributions are: *(i)* a novel paradigm for *semantic log analytics* that leverages knowledge-graphs to link and integrate heterogeneous log data; *(ii)* a *framework to generate RDF from arbitrary unstructured log data* through automatically generated extraction and graph modelling templates; *(iii)* a *set of base mappings, extraction templates, and a high-level general conceptualization* of the log domain derived from an existing standard as well as vocabularies for describing extraction templates; *(iv)* a prototypical *implementation* of the proposed approach, including detailed documentation to facilitate its reuse, and *(v)* an evaluation based on a realistic, multi-day log dataset [27]. All referenced resources, including the developed vocabularies[3], source code[4], data, and examples, are available from the project website[5].

The remainder of this paper is organized as follows: we introduce our log KG building approach in Section 2 and evaluate it in Section 3 by means of example

---

[3] `https://w3id.org/sepses/ns/log#`

[4] `https://github.com/sepses/slogert/`

[5] `https://sepses.ifs.tuwien.ac.at/`

use cases. We then contrast the appraoch against the state of practice (Section 4) and review various strands of related work (Section 5); finally, we conclude in Section 6 with an outlook on future work.

## 2   Building knowledge graphs from log files

In this section, we introduce the SLOGERT (Semantic LOG ExtRaction Templating) log KG generation framework and discuss its architecture, components, and their implementation. The associated workflow, illustrated in Figure 1, transforms and integrates arbitrary log files provided as input. It consists of two major phases: (1) template extraction, which results in an RDF pattern for each type of log message that appears in the sources; and (2) graph building, which – based on these patterns – transforms raw log data into RDF.[6]

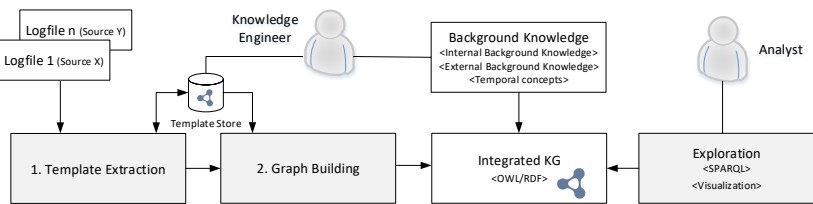

Fig. 1: SLOGERT workflow

In the *Template Extraction* phase, SLOGERT will automatically generate event templates from unstructured log data by identifying the different types of log messages and their variable parts (i.e., parameters) in the raw log messages. For this, we rely on a well-established log parser toolkit [46] that generates *extraction templates*, which at this stage do not provide any clues about the semantics of the log message or the contained parameters. To enrich these extraction templates with semantics, we next annotate the parameters (variable parts) according to their type as well as extract relevant keywords from the log messages, which are used to link each log template with relevant CEE [29] annotations.

We then use this information to associate each log extraction template with a corresponding *RDF graph modelling template* (represented as Reasonable Ontology Templates (OTTRs)). The resulting graph modelling templates can be annotated, adapted, extended and reused, i.e., they only have to be generated (and optionally extended) once from raw log data in which unknown log events appear.

In the *Graph Building* phase, SLOGERT then parses each line in a log file and applies the matching extraction and RDF modelling templates to transform

---

[6] A more detailed documentation and pseudocode specification of the process is available at `https://github.com/sepses/slogert/`.

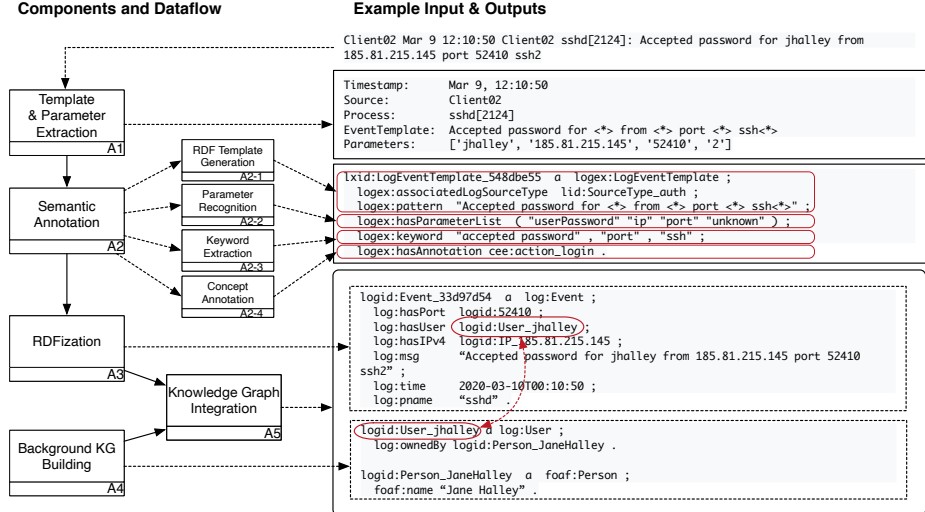

Fig. 2: SLOGERT components and processing of a single example log line

them into RDF. Thereby, we generate entities from textual parameters in the log stream and represent them in our log vocabulary. Combining the generated RDF from multiple, potentially heterogeneous log files and log sources results in a single integrated log KG. This graph can contextualize the log data by linking it to existing background knowledge – such as internal information on assets and users or external information, e.g., on software, services, threats etc.

Finally, analysts can *explore*, query, analyze, and visualize the resulting log KG seamlessly across log sources.

### 2.1  SLOGERT Components

Following this high-level outline of the SLOGERT workflow, this section describes each component in more detail. For a dynamic illustration of the overall process by way of an example log line, cf. Figure 2.

### Phase 1: Template Extraction

*Template & Parameter Extraction (A1),* i.e., the first step in the process from raw log lines to RDF, relies on LogPAI[7] [46], a log parsing toolkit that identifies constant strings and variable values in the free-text message content. This step results in two files, i.e., *(i)* a list of log templates discovered in the log file, each including markings of the position of variable parts (parameters), and *(ii)* the actual instance content of the logs, with each log line linked to one of the log template ids, and the extracted instance parameters as an ordered list.

---

[7] https://github.com/logpai/logparser

This process is fully automated and applicable to any log source, but depending on the structure of the log messages, it may not necessarily result in clearly separated parameters. As an example, consider that a user name next to an IP address will be identified as a single string parameter, as they usually change together in each log line. To achieve better results, LogPAI therefore accepts regular expression specifications of patterns that should be extracted as parameters, if detected. We take advantage of this capability by defining general regex patterns for common elements and including them in the configuration. At the end of this stage, we have *extraction templates* and the associated extracted instance data, but their semantic meaning is yet undefined.

*Semantic Annotation (A2)* takes the log templates and the instance data with the extracted parameters as input and *(i)* generates RDF rewriting templates that conform to an ontology and persists the templates in RDF for later reuse (A2-1), *(ii)* detects (where possible) the semantic types of the extracted parameters (A2-2), *(iii)* enriches the templates with extracted keywords (A2-3), and *(iv)* annotate the templates with CEE terms (A2-4)

For the parameter type detection (A2-1), we first select a set of log lines for each template (default: 3) and then apply rule-based Named Entity Recognition (NER) techniques. Specifically, we use TokensRegex from Stanford CoreNLP [5] to define patterns over text (sequences of tokens), and map them to semantic objects. CoreNLP can detect words in combination with part of speech (POS) tags and named entity labels as part of the patterns.

Such token-based extraction works well for finding patterns in natural language texts, but log messages often do not follow the grammatical rules of typical natural language expressions and contain "unusual" entities such as URLs, identifiers, and configurations. For those cases, we additionally apply standard regex patterns on the complete message. For each identified parameter, we also define a type and a property from a log vocabulary to use for the detected entities. In case a parameter does not result in any matches, we mark it as unknown.

In our prototype, we collect all parameter extraction patterns in a YAML configuration file and model a set of generic patterns that cover various applications, including the illustrative use cases in Section 3. These patterns are reusable across heterogeneous log sources and can be easily extended, e.g., with existing regex log patterns such as, e.g., Grok[8]. For the semantic representation necessary to allow for a consistent representation over heterogeneous log files, we followed the Ontology 101 methodology [33], extended a prior log vocabulary [26] and mapped it to the Common Event Expression (CEE) [29] taxonomy. Furthermore, we persist our ontology with a W3ID namespace (i.e., `https://w3id.org/sepses/ns/log#`) and use Widoco [14] for the ontology documentation.

Our vocabulary core represents log events (`log:Event`) with a set of fields (sub-properties of the `log:hasParameter` object property and `log:parameter` datatype properties). Each log event originates from a specific host (`log:hasSource`

---

[8] `https://github.com/elastic/logstash/blob/v1.4.2/patterns/grok-patterns`

Host) and exists in a specific log source (`log:hasLogSource`, e.g., an FTP log file). The underlying source type (e.g., ftp) is expressed by `log:SourceType`, and the log format is represented as `log:Format` (e.g., syslog). Furthermore, a log event template is tagged with its underlying action (e.g., login, access), domain (e.g., app, device, host), object (e.g., email, app), service (e.g., auth, audit), status (e.g., failure, error), and a subject (e.g., user) based on the CEE specification[9].

Once we have identified extraction templates for events and parameters, we can generate corresponding RDF generation templates for each of them. This step expresses the patterns that determine how KGs are built from the log data as reusable OTTR [1] templates, i.e., in a language for ontology modeling patterns. As all the generated templates are reusable and should not have to be regenerated for each individual instance of a log file, we persist them in RDF with their associated hash (based on the static parts of the log messages) as identifier. Finally, as a prerequisite to generate the actual KG from these OTTR templates, we transform all log line instances of the input into the stOTTR format, a custom file format similar to Turtle, which references the generated OTTR templates.

**Phase 2: Graph Building** In this step, we generate a KG based on the OTTR templates and stOTTR instance files generated in the extraction component.

*RDFization (A3)* For the conversion of OTTR templates and instance files, we rely on Lutra[10], the reference implementation for the OTTR language. Thereby, we expand the log instance data into an RDF graph that conforms to the log vocabulary and contains the entities and log events of a single log file.

*Background KG Building (A4)* Linking log data to background knowledge through the use of appropriate identifiers is a key step that facilitates enrichment with both local context information and external knowledge. The former represents information that is created and maintained inside the organization and not intended for public release. Examples include, e.g., the network architecture, users, organizational structures, devices, servers, installed software, and documents.

This knowledge can either be maintained manually by knowledge engineers or automatically by importing, e.g., DHCP leases, user directories with metadata, or software asset information. The dynamic nature of such information (e.g., a user switches department, a computer is assigned a new IP address, software is uninstalled) necessitates a mechanism to capture temporal aspects. To this end, RDF-Star can be used to historize the contained knowledge.

The second category, external knowledge, links to any publicly available (RDF) data sources, such as, e.g., IT product and service inventories, vulnerability databases, and threat intelligence information (e.g., collected in [24]).

---

[9] `https://cee.mitre.org/language/1.0-beta1/core-profile.html`
[10] `https://ottr.xyz/#Lutra`

*Knowledge Graph Integration (A5)* combines the KGs from the previously isolated log files and sources into a single, linked representation. Key concepts and identifiers in the computer log domain follow a standardized structure (e.g., IP and MAC addresses, URLs) and hence can be merged using the same vocabulary. In case external knowledge does not align with the generated graphs (e.g., entity identifiers differ), an additional mapping step has to be conducted before merging. Existing approaches, such as the Linked Data Integration Framework Silk[11] can be used for this purpose.

## 3   Use Cases & Performance

We illustrate the presented approach and its applicability to real-world log data based on a systematically generated, publicly available data set that was collected from testbeds over the course of six days [27]. Furthermore, we report on the performance of the developed prototype (cf. Section 3.3).

### 3.1   Data Source

The AIT log dataset (V1.1)[12] contains six days of log data that was automatically generated in testbeds following a well-defined approach described in [27]. It is a rare example of a readily available realistic dataset that contains related log data from multiple systems in a network. In addition, information on the setup is provided, which can be used as background knowledge in our approach. As detailed information on the context of the scenario was not available, we complemented it with synthetic example background knowledge on the environment that the data was is generated in for demonstration purposes.

Each of the web servers runs Debian and a set of installed services such as Apache2, PHP7, Exim4, Horde, and Suricata. Furthermore, the data includes logs from 11 Ubuntu hosts on which user behavior was simulated. On each web server, the collected log sources include Apache access and error logs, syscall logs from the Linux audit daemon, suricata logs, exim logs, auth logs, daemon logs, mail logs, syslogs, and user logs. The logs capture mostly normal user behavior; on the fifth day of the log collection (2020-03-04), however, two attacks were launched against each of the four web servers. In total, the data set amounts to 51.1 GB of raw log files.

### 3.2   Use Cases

In this scenario, we assume that activities have raised suspicion and an analyst wants to conduct a forensic analysis based on the available log data. We will illustrate how our proposed framework can assist in this process. To this end, we first processed all raw logs[13] with SLOGERT and stored them together with

---

[11] http://silkframework.org/

[12] https://zenodo.org/record/4264796

[13] From the audit logs, we only extracted the time frame relevant for the investigation.

| template | timestamp | host | sourceType | annotations |
|----------|-----------|------|------------|-------------|
| 108cf6f8 | 2020-03-04T19:26:00 | mail.cup.com | messages | failure,login |
| ... | ... | ... | ... |
| 108cf6f8 | 2020-03-04T19:28:59 | mail.cup.com | messages | failure,login |
| 108cf6f8 | 2020-03-04T19:29:00 | mail.cup.com | messages | failure,login |
| c9f3df73 | 2020-03-04T19:29:07 | mail.cup.com | syslog | login,success |

Table 1: Query result for activities of a given user in the network (excerpt)

the background knowledge in a triple store[14]. Overall, we collected 838.19MB of raw log files, resulting in 84,827,361 triples for this scenario.

```
PREFIX log: <https://w3id.org/sepses/ns/log#>
PREFIX logex: <https://w3id.org/sepses/ns/logex#>
PREFIX rdfs: <http://www.w3.org/2000/01/rdf-schema#>
PREFIX xsd: <http://www.w3.org/2001/XMLSchema#>

SELECT ?template ?timestamp ?host ?sourceType ?annotations
WHERE {
        ?logEvent a log:Event ;
           log:time ?timestamp ;
           log:hasSource ?source ;
           logex:template ?templateId ;
           log:hasSourceHost / log:host ?host .
    ?templateId rdfs:label ?template .
    ?source log:hasSourceType / rdfs:label ?sourceType .
        ?logEvent log:hasUser / log:user.name "daryl" .
    FILTER (xsd:dateTime(?timestamp) > "2020-03-04T18:30:00"^^xsd:dateTime)
    OPTIONAL {
        { select ?templateId (group_concat(?anno;separator=',') as ?annotations) where {
                ?templateId a logex:LogEventTemplate ;
                    logex:hasAnnotation/rdfs:label ?anno
          } group by ?templateId } }
} ORDER BY ?timestamp
```

Listing 1: SPARQL query to show activities of a user

Listing 1 demonstrates how an analyst can query the activities associated with a given username (i.e., `daryl`). The query illustrates the ability to access integrated log data and the flexibility of SPARQL as a query language for log data analytics.

Table 1 shows an excerpt of the query results, with the template name, timestamp of the event, host, type of log, and automatically extracted CEE [29] annotation labels. The template associated with each log event makes it possible to easily identify events of the same type; human-readable labels can optionally be assigned in the template library.

As a simple illustrative example, the query makes use of only a small subset of the available extracted properties. To explore the context and increase an analyst's understanding of the situation in the course of an investigation, other extracted properties such as IP addresses, processes, commands, files, URLs, and email addresses can be added. These extracted entities establish links among

---

[14] GraphDB 9.5, `https://graphdb.ontotext.com`

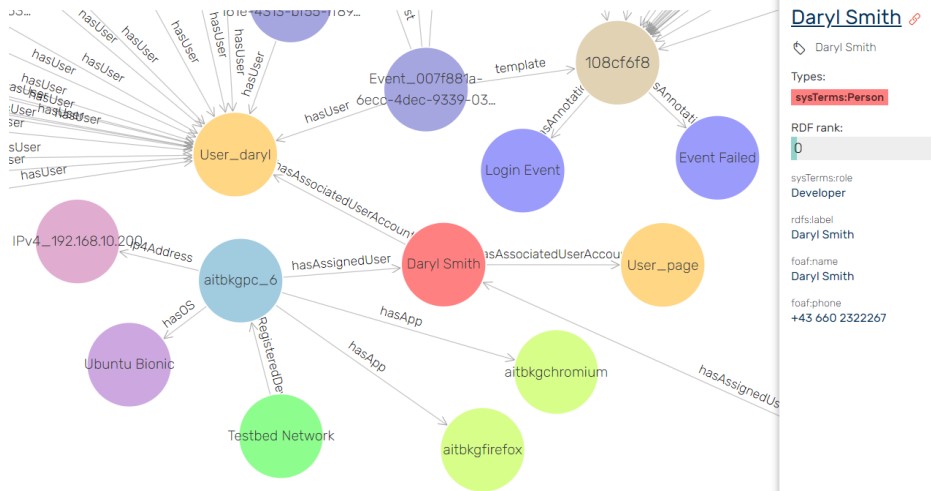

Fig. 3: Graph exploration: context information for `User_daryl`

log events and connect them to background knowledge, allowing the analyst to explore the log data from multiple perspectives in order to "connect the dots", e.g., in the context of attacks.

As an example, consider that the sequence of login failures in a short time period evident in Table 1 suggests a possible brute force attack, and the successful login shortly thereafter is alarming. To explore this further, the analyst can construct an enriched view on the available log events by visualizing the data (e.g., using GraphDB) and interactively following links of interest. The graph structure makes it possible to navigate the contextualized log data with interlinked background knowledge that otherwise is typically stored in external documentation or only exists in analysts' heads. In the example in Figure 3, the analyst started from the username `daryl` and navigated to the Person associated with the account; from there, she can obtain additional information about the person, such as the role in the company, contact information, assigned devices, and additional usernames (e.g., `User_page`). In a similar manner, the analyst could integrate and explore external RDF data sources. We can also see how events are connected to the user in the selected time frame, and how the templates are annotated with detected CEE terms.

### 3.3   Performance

For each log source in the AIT log dataset – collected from four servers in a testbed environment – we ingest the raw log files[15] and execute all steps in the workflow according to Figure 2. As the graph generation is split into multiple

---

[15] Note that we only used a relevant subset of the high-volume audit logs, but the full time range for all other logs.

| Source | Raw (MB) | TTL (MB) | HDT (MB) | P1 (s) | P2 (s) |
|---|---|---|---|---|---|
| access | 131.80 | 526.08 | 87.57 | 336 | 1398 |
| audit | 525.41 | 2026.38 | 305.45 | 6652 | 6799 |
| auth | 0.48 | 3.35 | 0.34 | 3 | 5 |
| daemon | 0.58 | 3.94 | 0.63 | 5 | 6 |
| error | 0.17 | 0.69 | 0.15 | 3 | 2 |
| fast | 3.15 | 15.89 | 2.34 | 5 | 34 |
| mail | 58.27 | 295.36 | 39.52 | 146 | 690 |
| main | 1.96 | 13.81 | 3.02 | 6 | 20 |
| messages | 28.60 | 108.10 | 12.55 | 34 | 199 |
| sys | 87.78 | 408.50 | 53.05 | 297 | 909 |
| Total/Merged | 838.19 | 3402.11 | 498.36 | 7486 | 10062 |

Table 2: Log sources and graph output. The run time for the following phases is measured in seconds: template extraction (P1) and graph building (P2)

processing steps, we measure the execution time for each phase (i.e., template extraction and graph building) on a single machine with a Ryzen 7 3700X processor (64GB RAM).

Table 2 shows the input log sources with their total file sizes over all four servers. Furthermore, it lists the sizes of the generated (intermediate) KGs in Turtle format (TTL), as well as in compressed format (HDT). At the end of the SLOGERT process, all intermediate graphs are merged into a single KG. In our illustrative scenario, we processed 838.19MB of raw log data in total and generated a KG from it that is 498.36MB in HDT format. Note that whereas the resulting TTL graph files were approx. four times the raw input size, the compressed HDT data, which can also easily be queried with SPARQL, is about 40% smaller than the original log file. The size of the generated graphs could further be reduced by *(i)* not including the full original raw input message (currently we keep it as message literal), and/or *(ii)* discarding unknown parameters, and/or *(iii)* extracting only the specific classes and properties necessary for a given set of analytic tasks.

In terms of processing time graph building (P2) is the most time consuming phase (approx. 168 min)[16]. We see that run-time scales linearly with file size; it can easily be reduced by parallelizing the semantic annotation and graph generation phases. Our prototype converts the log events into batches – the number of log lines per batch (200k in our experiments) can be configured. Although we executed the process on a single machine in sequence, each batch file could easily be processed in parallel. Taking the audit log as example, all audit log event lines were split into 200k batches (13 files), taking approx. 17 min each to build the KG.

---

[16] The Lutra team is working on performance according to their release notes.

| Aspect | Existing LMSs | SLOGERT |
|---|---|---|
| Mode | Online & Offline | (Currently) offline |
| Storage | Proprietary databases | RDF & HDT compression |
| Extraction | regex filters | template-based |
| Normalization | fields | entity hierarchies and links |
| Background knowledge | - | RDF linking |
| Data insights | Dashboards and Reports | Graph queries and navigation |

Table 3: Comparison of SLOGERT with existing LMSs

## 4   State of the Practice

Commercially available Log Management Systems (LMSs) – such as Splunk[17], Graylog[18], or Logstash[19] – prioritize aggregation, normalization, and storage over integration, contextualization, linking, enrichment, and rich querying capabilities. They are typically designed to allow scalable retention of large log data and focus on reporting and alerting based on relatively simple rules.

Table 3 compares and contrasts SLOGERT with such existing LMSs. Whereas some tasks, such as *log collection* from raw logs, can rely on available standard mechanisms, the approach differs fundamentally in terms of *event and parameter detection*, *normalization*, *background knowledge linking*, and the way that *insights* can be gained.

In particular, SLOGERT *(i)* automatically classifies events and assigns types from a taxonomy based on the static parts of the messages and *(ii)* identifies and annotates variable parts of the messages. Although existing LMSs often also include a predefined and limited set of extractors to identify relevant patterns (e.g., IP address, date/time, protocol), they do not capture entities, their relationships, and the nature of these relationships in a graph structure. Furthermore, they are limited in their representational flexibility by the structure of the underlying, typically relational, storage.

The graph-based approach makes it possible to link assets to concepts and instances defined in background knowledge in order to enrich log events with additional internal or external knowledge. For instance, multiple usernames can be linked to the same person they belong to or software assets can be linked to public sources such as the Cyber-security Knowledge Graph (CSKG) [24] to include information about their vulnerabilities.

Finally, whereas existing LMSs typically provide relatively static dashboards and reports, SLOGERT opens up possibilities for exploration through graph queries and visual navigation, providing a new flexible perspective on events, their context, and their relationships.

Overall, no comparable graph-based semantic systems exist – current state-of-the-art message-centric log management systems focus on on aggregation, management, storage, and manual step-by-step textual search and interpreta-

---

[17] https://splunk.com

[18] https://www.graylog.org/

[19] https://logstash.net

tion with implicit background knowledge. SLOGERT makes it possible to automatically contextualize, link and interpret log events. It complements, but does not replace established log management solutions with additional techniques to extract, enrich, and explore log event data. Specifically, our current focus is facilitating deeper inspection of subsets of log data from selected log sources, e.g., in a relevant time frame. To this end, we have developed flexible mechanisms for fully automated ad-hoc extraction and integration.

## 5    Related Work

*Log parsing and extraction* Logs, i.e., records of the events occurring within an organization's systems and networks [21], are composed of log entries, each of which provides information related to a specific event that has occurred. These log entries typically consist of a structured part with fields such as a timestamp, severity level etc., and an unstructured message field. Whereas conventions for the structured parts are somewhat standardized (e.g., in [2])), there is little uniformity on the content of the message field, despite numerous standardization attempts (e.g., IDMEF [18], CEE [29], CIM [9] and CADF [8]). Because log messages are produced from statements inserted into the code, they often do not follow typical natural language grammar and expression, but are shaped according to the code that generates them. Specifically, each log message can be divided into two parts: a constant part (i.e., fixed text that remains the same for every event instance) and a variable part that carries the runtime information of interest, such as parameter values (e.g., IP addresses and ports).

Traditional manual methods for analyzing such heterogeneous log data have become exceedingly labor-intensive and error-prone [15]. Furthermore, the heavy reliance on regular expressions in log management results in complex configurations with customized rules that are difficult to maintain as systems evolve [15]. These limitations of regex-based event extraction motivated the development of data-driven approaches for automated log parsing (e.g., [42]) that leverage historical log messages to train statistical models for event extraction. [15] provides a systematic evaluation of the state-of-the-art of such automated event extraction methods. We leverage these methods as the first step in our automated KG construction workflow. Specifically, our template extraction is based on the Log-PAI logparser toolkit [47], which provides implementations of various automated event extraction methods.

*Log representation in Knowledge Graphs* has attracted recent research interest because graph-based models provide a concise and intuitive abstraction for the log domain that can represent various types of relations flexibly. Therefore, a variety of approaches that apply graph-based models and graph-theoretical analysis techniques to log data have been proposed in the literature, covering applications such as query log analysis [10,45], network traffic and forensic analysis [43,7], and security [36]. Whereas these contributions are focused on graph-theoretical metrics and methods, another stream of knowledge-graph-centric literature has

emerged more recently. CyGraph [32], e.g., develops a property graph-based cybersecurity model and a domain-specific query language for expressing graph patterns of interest. It correlates intrusion alerts to vulnerability paths, but compared to our approach, it does not aim for semantic lifting of general log data.

In terms of semantic KGs, existing approaches have focused either on structured log data only [31], or on tasks such as entity [20] and relation [37] extraction in unstructured log data. Whereas some of the extraction methods introduced in this context are similar to our approach, their focus is less on log representation, but on cybersecurity information more general (e.g., textual descriptions of attacks).

Other contributions have focused on a conceptualization of the log domain and the development of appropriate vocabularies for log representation in KGs [13]. Another recent, more narrowly focused approach [26] that does not cover general extraction introduces a vocabulary and architecture to collect, extract, and link heterogeneous low-level file access events from Linux and Windows event logs. Finally, [24] provides a continuously updated resource that links and integrates cybersecurity information, e.g., on vulnerabilities, weaknesses, and attack patterns, providing a useful linking target in the context of this log extraction framework.

## 6    Conclusions and Future Work

This paper introduced SLOGERT, a flexible framework and workflow for automated KG construction from unstructured log data. The proposed workflow can be applied to arbitrary log data with unstructured messages and consists of a template extraction and a graph building phase. Our prototype demonstrated the viability of the approach, particularly if the messages in the log sources do not require frequent relearning of the extraction templates. Configurability and extensibility were key design goals in the development of SLOGERT. For arbitrary log data with unstructured messages in a given log domain, the framework generates a keyword-annotated RDF representation. The demonstrated configuration covers standard concepts for various log sources relevant in a cybersecurity context, however, they can easily be adapted for different log domains.

An inherent limitation evident from our experiments was a sensitivity to the training data set during template extraction; specifically, entities can not be properly identified if there is too little variation in the variable parts of a given log message. This limitation can be tackled through larger log collections, ideally through a community effort towards creating a shared library of extraction templates for standard log data sources.[20] More broadly, we also envisage a community-based effort to develop mappings, extensions for specific log sources, and shared domain knowledge such as vulnerability information and threat intelligence.

---

[20] Note that the template representation in RDF simplifies sharing.

Due to the widespread use of unstructured log data in numerous domains and the limitations of existing analytic processes, we expect strong adoption potential for SLOGERT, which in turn could also drive adoption of Semantic Web technologies in log analytics more generally. Furthermore, we also expect impulses for KG research and takeup by KG builders that need to integrate log data into their graphs.

In our own research, we will apply the proposed approach in the context of semantic security analytics[21]. Our immediate future work will focus on the integration into logging infrastructures, e.g., by supporting additional formats and protocols. Furthermore, we will focus on graph management for template evolution and incremental updating of log KGs.

Conceptually, our bottom-up extraction approach provides a foundation for future work on linking it to higher-level conceptualizations of specific log domains (e.g., based on DMTF's CIM [9] or the CADF [8] event model). Potentially, this can also provide a foundation for research into event abstraction, i.e., automatically transforming a sequence of individual log events into higher-level composite events or log-based anomaly detection, e.g., through combinations of rule-based methods and relational learning and KG embedding techniques.

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
