# OpenReview forum: "The SLOGERT Framework for Automated Log Knowledge Graph Construction"
_eswc-conferences.org/ESWC/2021/Conference/Resources_Track — ESWC 2021 Resources_

### Official Review · AnonReviewer5 · 2021-01-11
**Review for The SLOGERT Framework**

**Rating:** 2
**Confidence:** 4

**Review:**

### After Rebuttal

The answer of the authors provide clarifications for all of my concerns. However, I encourage the authors to:

- Put the algorithm of "template enrichment" in the body of the paper, or at least as an appendix in an extended version. To have this algorithm improves the quality of your work.
- Put the discussion of applicability to arbitrary logs and the parts developed by you in the body of the paper, since this discussions clarify your contribution.

Thus, I change my review to accept.

# General comments

The authors present SLOGERT, which is a framework to create knowledge graphs from log files. I think that the topic is very interesting and also that this tool may be useful in practice, principally for visualizing log data. However, there are some points that make me not be clear about my decision.

# Detailed comments
The paper starts with a detailed introduction and a depth revision of the related work. However, I think that the content of Table 1 should be explained more in depth, to understand the benefit of using the approach proposed by the authors. At section 3 the authors explain the workflow of their approach, which can be well understood at a high level.

However I was expecting a formal definition of the algorithm that obtains the templates from the log files. Also, it would be interesting to have the algorithm for the graph construction (in an algorithmic environment).

At some point the authors declare that "This process is fully automated and applicable to any log source, but depending on the structure of the log messages, it may not necessarily result in clearly separated parameters''. Thus, I have some concerns about to what extent this approach is applicable in practice, since for me it is not clear how your approach can get the different templates. Are you able to handle an arbitrary log file, or you have to put some constraints in order to parse a log file?

A second concern is that I would like to know which parts of your implementation rely on libraries and which parts are developed by you. I have the feeling that a great part of your approach depends on external libraries. In order for this paper to be accepted, I think that it is important to understand your contributions; and again, I think that including the definition of the algorithms for getting the templates and for constructing the graph can be useful to understand the contributions.

The use case is interesting, and as expected, it is possible to see a visualization of a log file. Also, I think that the performance is good enough for this engine. However, I missed a comparison with respect to an existing approach, but this is not crucial for me.

Hence, I think that the proposal of the authors is interesting and it can have many use cases in practice. However, I'm a bit concerned about the extendability of this approach for arbitrary log files. And in terms of the paper, I think that it is a good idea to present the algorithms that are being used for the template extraction and the graph construction; moreover, this could be interesting to understand the components developed by you and the ones that rely on libraries.

Thus, for me this is a borderline paper, but my opinion may change depending on the answer of the authors.

# Other comments

- The footnote 3 and 5 are the same.
- The font in Figure 2 is too small. It is difficult to read.
- I disagree with the usage of Python 2 in your implementation. In order for this approach to be used in practice you should migrate to Python 3.
- You should explain the domain of the data for the use case in the paper. I had to go to the page to get this.

**Anonymity:**

Yes, I would like my review to remain anonymous.

**Strong Points:**

- The paper presents a novel technique that has a lot of potential.
- The discussion of the related work and the motivation of the problem.
- The analysis of an interesting use case.
- The reproducibility of their experiments.

**Subreviewer:**

I submitted this review.

**Weak Points:**

- The lack of a formal definition of their algorithms.
- I'm not sure about which parts of their approach was developed by the authors and which parts depend on external libraries.
- The lack of a comparison with respect to an existing approach.

---

> ### Author Rebuttal · Authors · 2021-01-29
>
> Thank you for the constructive feedback and questions, to which we would like to respond as follows:
>
> ### Definition of algorithms
> To illustrate the process in more detail, we have extended Fig. 1, which can be found in the repository cf. [1]. In the following, we outline the algorithm for the template enrichment, which is a core part of our contribution:
>
> #### [A2 - Template Enrichment]
>
> *  Load existing `RDF_templates` list
> *  Load `regex_patterns` from `config` list for parameter recognition
> *  Initialize `NLP_engine` engine
> *  For each `extraction-template` from the list of `<extraction-template, raw-result>` pairs
> 	* Transform `extraction-template` into an `RDF_template_candidate`
> 	* if `RDF_templates` do not contain `RDF_template_candidate `
> 		* **[A2.1 - RDF template generation]**
>           * For each `parameter` from `RDF_template_candidate`
>               * If `parameter` is `unknown`
>                   * **[A2.2 - Template parameter recognition]**
>                     *  Load `sample-raw-results` from `raw-results`
>                     *  Recognize `parameter` from `sample-raw-results` using `NLP_engine` and `regex_patterns` as `parameter_type`
>                     *  Save `parameter_type` in `RDF_template_candidate`
>                   * **[A2.2 - end]**
>           * **[A2.3 - Keyword extraction]**
>             * Extract `template_pattern` from `RDF_template_candidate`
>             * Execute `NLP_engine` engine on the `template_pattern` to retrieve `template_keywords`
>             * Add `template_keywords` as keywords in `RDF_template_candidate`
>           * **[A2.3 - end]**
>           * **[A2.4 - Concept annotation]**
>             * Load `concept_model` containing relevant concept in the domain
>             * For each `keyword` from `template_keywords `
>                 * for each `concept` in `concept_model`
>                     * if `keyword` contains `concept`
>                         * Add `concept ` as concept annotation in `RDF_template_candidate`
>           * **[A2.4 - end]**
>           * add `RDF_template_candidate` to `RDF_templates` list
> 		* **[A2.1 - end]**
>
> A detailed overview on the complete process can be found at [2].
>
> ### Applicability to arbitrary log files
> Configurability and extensibility was a key design goal in the development of SLOGERT. In general, the framework can transform any arbitrary log files with unstructured messages into RDF. For a new log domain, this will at a minimum result in a keyword-annotated rdf representation. To extract domain-specific concepts, it is necessary to annotate the automatically identified variable parts in log message patterns. The standard configuration provided (i.e., regex patterns, annotated templates, log extraction settings etc.) covers standard concepts for various log sources relevant in a cybersecurity context for illustration purposes, but they can easily be adapted for different log domains
> We have extended the documentation of the configuration of the framework on github, which should make it easy to configure the framework for new sources [2]. We also plan to run tutorials on SLOGERT and semantic log analysis at conferences.
>
> ### Which parts were developed by the authors and which parts depend on external libraries
>
> SLOGERT is a novel framework for log KG generation which uses semantic web libraries for certain steps. We provide an updated version of Fig. 2 [1] to more clearly illustrate the process (also cf. [2]). Based on this figure, we can explain this as follows:
>
> * A1 Template and parameter extraction is handled by an existing library (LogPAI) that we integrate into SLOGERT.
> * A2 Template annotation, including RDF template generation, parameter recognition, keyword extraction, and concept annotation are a core development artifact; it only relies on CoreNLP for the keyword annotation.
> * A3 uses OTTR as a flexible RDF generation engine
> * A4 is user-contributed and application-specific
>
> ### Comparison with respect to an existing approach
>
> Completely automating the generation of knowledge graphs from log data has, to the best of our knowledge, hitherto been an unsolved problem. In the section “State of the Practice”, we contrast the approach to available log management systems.
> The use case visualization does not only visualize the log file but also shows links to internal knowledge (e.g., user information and device information), which is typically not available in log tools and not directly connected. Hence, the resulting knowledge graphs connect data sources outside of the log file, which is one of the benefits.
>
> ### Other comments
> Thank you for the comments, we fixed them in the paper. As other reviewers also recommended, we will move the related work section towards the end of the paper, and remove the line pointing to future work from Table 1.
>
> [1] https://raw.githubusercontent.com/sepses/slogert/master/slogert.jpg
> [2] https://github.com/sepses/slogert/blob/master/README.md

---

> > ### Comment · AnonReviewer5 · 2021-02-03
> > **Answer**
> >
> > Thank you for your answer. Most of my concerns are clarified. However, as I put in the edited review, it is important to add certain discussions to the body of the paper. Also, it is important to have access to the algorithm (at least in an appendix) not only explained with a figure, but defined formally too (with an algorithmic environment in Latex).

---

### Official Review · AnonReviewer2 · 2021-01-14
**An emerging and promising software resource which requires more maturity**

**Confidence:** 4

**Review:**

====
I acknowledge the comments and clarifications provided by the authors. They have made sincere efforts to address concerns via actions such as providing an updated figure 1 as well as improvements to the documentation. I am still not completely convinced about the impact and reusability of this resource especially for users/practitioners outside the Semantic Web community where this tool is going to be most impactful. In my opinion, a few additional APIs and little more focus on usability along with a roadmap for adoption and maintenance will go a long way in making the paper and more importantly the resource strong. I am still on the borderline for the paper.
====

This paper presents a framework for ingesting and mapping data in log files (e.g. software system logs) into a knowledge graph (KG). From a resource point of view, the paper contributes a research software prototype (called SLOGERT) for automatically ingesting log files into a KG.

**Impact:** Limited research has focused on mapping log file data into knowledge graphs and SLOGERT extends some of this existing work by automatically inferring the templates of log files with the help another existing resource (LOGPAI) before mapping the data into RDF. SLOGERT maybe of minimal interest within the Semantic Web community, but can be potentially useful to researchers and developers in the security community. A variety of downstream system security applications can be enabled on top of knowledge graph populated from heterogeneous log files.

**Reusability:** SLOGERT as a resource seems to be fairly new with little to no evidence of its adoption beyond its creators. Minimal documentation is provided to help run the prototype. Beyond the paper, there are no additional tutorials on how to leverage or extend the tool. For instance, the instructions on running SLOGERT talks about setting properties in the config file, but no instructions are provided on the definition/meaning of the properties/keys or their acceptable values. Another place where installation/running instructions could be improved is by including the required python packages into a requirements.txt file.

**Technical Quality:** The approach presented in the paper is technically sound. SLOGERT reuses several high quality resources (e.g. LOGPAI, Stanford CoreNLP) as part of its solution. Overall, it's easy to follow the approach presented in the paper. Details about the Semantic Annotation (A2) approach could be improved. It's unclear how keywords from the messages are extracted and then mapped in to the CEE. Is it performed via regular expression/pattern matching with the help of CoreNLP? If yes, what are some of these example patterns? Along with the run time and size metrics, a "correctness" or "accuracy" metric could also be considered. Accuracy could be measured at different levels, e.g. a) was the log line mapped to the correct template b) were all the appropriate keywords extracted c) was the CEE mapping accurate.

**Availability:** SLOGERT's source code is available on GitHub with an MIT license. No maintenance/sustainability plan was available or discussed.

My overall impression is that this a research prototype and needs a little more maturity before it could be adopted by researchers/developers in other communities. SLOGERT so far has been evaluated on about 4-5 log file types. For this tool to make impact in a non-Semantic Web community, authors would need to make it easier for users to ingest the generated triples into a triple store and also provide user-friendly APIs or mechanisms to interact with the data.


**Anonymity:**

Yes, I would like my review to remain anonymous.

**Rating:**

-1: Weak Reject

**Strong Points:**

SLOGERT leverages several existing high quality resources as part of its solution. Automatically inferring the templates of log files with LOGPAI is a very handy feature.

**Subreviewer:**

I submitted this review.

**Weak Points:**

From a resource point of view, SLOGERT is a research prototype that requires more maturity before it could be adopted and achieves the promised potential impact. Documentation and its usability for users outside the Semantic Web community needs to be improved.

---

> ### Author Rebuttal · Authors · 2021-01-29
>
> Thank you for the constructive feedback and questions, to which we would like to respond as follows:
>
> ### Reusability and Impact
> The resource is aimed to be useful for both, domain-specific communities that have to deal with log-structured data and for the semantic web community. For the latter, we also expect SLOGERT to become a valuable resource that opens up log-structured data sources and allows researchers and practitioners to easily integrate such data, thus lowering the barrier to leverage such data.
>
> The second target group, i.e., domain experts and analysts from other domains, includes security analysts but is not limited to this group. While we chose a log data set from this domain as a motivating example, the framework is not restricted to security use cases. As an example, during the development of the idea, we also discussed the approach with DevOps engineers handling large microservices log data which often requires infrastructure knowledge that is typically not directly available in the log analysis tools and cannot easily be linked and integrated. We agree with the reviewer that adoption will be more difficult for such audiences,  but we have recently seen a rise in interest in graph-based log-analytic techniques and see this as an opportunity to introduce these communities to Semantic Web technologies. Specifically, we plan to run SLOGERT-tutorials designed for uninitiated audiences.
>
> Regarding the comment that “it should be easier for users to ingest the generated triples into a triple store and provide user-friendly APIs or mechanisms to interact with the data”: the triples generated by SLOGERT can easily be ingested into a triple store and we also currently consider extending the approach to allow for continuous streaming updates. Complementing SLOGERT with APIs and (domain-specific) log-analytic tools that allow users not familiar with SPARQL to interact with the extracted data are exciting directions for future work that we follow as part of our research project SEPSES, which aims to develop a larger stack for semantic security analysis.
>
> ### Technical quality
> Following the recommendation to improve the explanation of the semantic annotation approach in step A2 and the sub-steps (i.e., RDF template generation, parameter recognition, keyword extraction and concept annotation), we have extended Fig. 1 and will include a new version (cf. [1]) that illustrates the process in more detail in the final paper.
>
> Regarding the question how keywords are mapped to CEE, this is done in a two-step process: the keywords are indeed extracted with CoreNLP; the mapping to CEE concepts is - in the current implementation - done through simple string matching between the keyword and the labels of the CEE concepts.  Note that using multiple labels per concept is possible (e.g., to map the keyword “accepted password” to “cee:action_login”). We will add a clarification on that in the paper. FWIW, we also consider other options such as mapping configurations, distance metrics etc., but leave this to future work.
>
> Regarding the comment that a “correctness” or “accuracy” metric could be considered, we can report on accuracy in terms of correctly mapped templates and the number of correctly extracted parameter types in our use case.
> Note that we did not strive to achieve completeness for a specific data set or log data source in our evaluation use case, but added generic patterns [2] to demonstrate the general usefulness. However, the configurable framework allows users to edit all aspects of the identified templates and the accuracy can therefore be improved in case of incorrectly mapped templates, incomplete or inappropriate keywords, or inaccurate CEE mappings.
> It will also generally be necessary to adapt them for complete coverage in different log domains.
>
> ## Documentation
> We have extended the documentation of the configuration of the framework on github, which should make it easy to configure the framework for new sources [4] for users inside and outside the Semantic Web community. We also plan to run tutorials on SLOGERT and semantic log analysis at conferences.
>
> [1] https://raw.githubusercontent.com/sepses/slogert/master/slogert.jpg
> [2] https://github.com/sepses/slogert/blob/master/src/main/resources/config.yaml
> [3] http://w3id.org/sepses
> [4] https://github.com/sepses/slogert/blob/master/README.md

---

> > ### Comment · AnonReviewer2 · 2021-02-04
> > **Thank you.**
> >
> > Thank you for your comments and clarifications.

---

### Official Review · AnonReviewer3 · 2021-01-15
**Review for The SLOGERT Framework for automated Log Knowledge Graph Construction**

**Confidence:** 4

**Review:**

The authors present an approach for turning log files into knowledge graphs such that the logs can get analysed using SPARQL.
Their approach consists in a straight-forward pipeline that 1. extracts templates for the log files, 2. applies those templates to produce RDF from log files, and 3. enriches that data with other data. The actions done in the 2-3 steps of the pipeline are straight-forward applications of existing tools (logpai, ottr, lutra, silk). The authors survey related work from log parsing and logs+Knowledge Graphs. Last, they present a case study with performance metrics.

## after rebuttal:
I have read the other reviews and the extensive rebuttals. Thanks to the authors! I remain sceptical about the paper. Agreeing with some of the other reviewers, I think the CfP demands more maturity and re-use and downgrade my review. I stand by my remark regarding (little) originality.

**Anonymity:**

Yes, I would like my review to remain anonymous.

**Rating:**

-1: Weak Reject

**Strong Points:**

* Clarity: the paper is well written
* Potential impact and Significance: I think the direction of logs+KGs is an important one for the future
* Design & Technical quality: The authors did not re-invent the wheel and re-used other resources
* Availability: the resource is on GitHub under the MIT license

**Subreviewer:**

I submitted this review.

**Weak Points:**

* Reusability: The resource is available online. Logs are everywhere, so a lot of potential. Yet, the reusability criterion from the CfP mentions re-use by third parties for which the authors did not provide evidence.
* Originality: The main contribution of the paper, the pipeline is short (2-step), what somebody would do intuitively, and is merely an application of the state-of-the-art solution in each building block. In my opinion, this is a very low  "level of invention". This is my main weak point.
* Related work: As logs are inherently line-based I would consider approaches that allow for CSV treatment using SPARQL as related work. For instance, http://tarql.github.io/

---

> ### Author Rebuttal · Authors · 2021-01-29
>
> Thank you for the positive evaluation and for the constructive feedback and questions, to which we would like to respond as follows:
>
> It is correct that SLOGERT leverages and integrates proven standard libraries that have been designed and work well for specific tasks - specifically log parsing (LogPai), keyword extraction (Stanford NLP) and RDF templating (OTTR); however, we do not agree with the assessment that this reuse of “state-of-the-art solutions” in each building block, as pointed out by the reviewer, is a weakness. Completely automating the generation of knowledge graphs from log data has, to the best of our knowledge, hitherto been an unsolved problem and we do not agree that the result is “what somebody would do intuitively”. Overall, reuse of existing components as part of a novel framework does in our view not diminish SLOGERT’s contribution as an innovative resource that constructs KGs from log-structured data.
>
> Regarding the comment that the “pipeline is short (2-step)”, we would like to point out that the template extraction and graph building phases illustrated in the high-level overview in Fig. 1 involve a large number of steps. To lay these steps out more clearly, we have extended Fig. 2 to include more details on the full process and will include an updated version [1] in the paper; we will also - within the space constraints - update the associated explanation in the final version of the paper. This should help to highlight the contribution that goes beyond a straight-forward application of existing tools more clearly.
>
> Finally, we will also follow the recommendation to include approaches that allow to access tabular data using SPARQL in the related work (although this solves only the relatively minor part of column mapping and does not address the more difficult message extraction).
>
> [1] https://raw.githubusercontent.com/sepses/slogert/master/slogert.jpg

---

### Official Review · AnonReviewer4 · 2021-01-17
**Nice approach for log mining**

**Rating:** 2
**Confidence:** 3

**Review:**

This paper describes SLOGERT, a framework for creating Knowledge Graphs out of log files. The authors provide an implementation of the approach, and illustrate it with an example processing 51GB of log files, which result in more than 84 million triples.

The paper is well written and relevant for the ESWC audience. The topic addressed in this work (i.e., extracting and structuring log files) is not novel, but the process of transforming these logs into KGs where more information is linked for analysts is very valuable. The authors also do a nice review of existing work for process mining, using many existing tools for their needs. Overall, I think the paper would be a nice contribution to the conference.

When doing my review, I noticed some points that needed clarification. I detail them below:

- The process of KG generation is quite obscure. The authors highlight different techniques that may be used to connect, link and enhance the resultant log KGs, but it's not clear whether these techniques have been used or how. This is also not shown in the illustrative example. There seem to be some activities that have to be maintained by knowledge engineers, but it's not clear how much effort or expertise is required vs what SLOGERT natively covers.

- The query shown in the example section doesn't show the benefits of using a KG versus a regular database. Since this is one of the benefits of this work versus other approaches, it would be nice to show the connection to external entities or terminology. For example, the authors mention a database of vulnerabilities. How would they be integrated in such a query?

- Has SLOGERT (or its results) been used by analysts? There are no reports on this in the paper, and I find hard to believe that analysts would be willing to learn how to perform complex SPARQL queries without an additional layer/API on top of the SPARQL endpoint.

- I am a little confused by some of the methodology proposed by the authors. On the one hand, existing work is criticized because maintaining regular expression is costly. At the same time the authors state that: "We take advantage of this capability by defining general regex patterns for common elements and including them in the configuration". I find this contradictory.

- I would place the state of the art at the end of the paper, as it is quite confusing to see a comparison of related work against SLOGERT when the framework has not yet been introduced. In addition, if something is future work for SLOGERT it should not be part of the comparison table. The other tools also could have many features under development, and yet they don't appear.

- There is a pointer to the code in the paper, but I haven't seen much information about how to configure the tool, or its examples. I think this should be provided if the authors aim for a wider adoption of SLOGERT.

- On a minor note, the W3C PROV ontology (https://www.w3.org/TR/prov-o/) was designed to model events (prov:Activity) such as the ones described here. It would be nice link the "log":" vocabulary to PROV.


**Anonymity:**

No, I would like my review to be deanonymized.

**Strong Points:**

- Usefulness of the approach, towards helping analysts.
- Ability to recognize log patterns and link them together in a KG

**Subreviewer:**

I submitted this review.

**Weak Points:**

[After rebuttal]: The authors have clarified the KG creation and tool reusability in their documentation
-----
- There is a sample evaluation, but the system does not illustrate whether it can help analysts or not
- The process for KG creation is obscure, I am not sure the approach would be reusable by others.

---

> ### Author Rebuttal · Authors · 2021-01-29
>
> Thank you for the positive evaluation and for the constructive feedback and questions.
>
> ### Benefits of KGs
> We agree that the benefits of the KG approach go well beyond what we could illustrate in the examples in the paper. Note, however, that we do show, for instance, (i) how log data is automatically integrated across hosts and log sources (cf. Table 2) and (ii) how internal background knowledge on users, devices and the networks can be navigated directly from interlinked log data (cf. Fig. 3). This is enabled by the shared vocabulary and RDF structure and unlike existing log management systems, where such information is typically not integrated, this allows to “connect the dots” when searching for clues in vast amounts of log data.
> Linking to external knowledge is another current line of work, e.g., to link software that appears in log files to vulnerability information from linked open data repositories, such as our Sepses KG [1]  or UCO: Unified Cybersecurity Ontology [2].
> Overall, the SLOGERT framework serves as a foundation and we plan to exploit the knowledge graphs generated by it in future work.
> ### Practical usefulness to security analysts
> SLOGERT has - at this stage - not yet been in production use; our current focus in this paper is on extraction and knowledge graph construction, which provides a foundation for future work on knowledge-driven security analytics. Specifically, SLOGERT is developed in the context of a larger toolstack developed with the SEPSES project for semantic security analysis. We did, however, use a comprehensive data set in the evaluation and performed an initial face validation with domain experts at WU and SBA Research, an industrial security research center involved in the project.
> Practical benefits pointed out were the ability to generate a knowledge graph that links isolated indicators from dispersed logs and contextualizes them, which contrasts favorably to current practice, which is typically a tedious manual process. The ability to visualize and navigate the logs was also considered particularly useful in demonstrations. We agree, however, that specialized APIs, analytic tools, and user interfaces on top of the framework would make it more accessible to analysts without experience or time to learn SPARQL. User-facing layers on top of the extraction framework are on our research agenda.
> ### Regular expressions
> Our criticism of current practice is that manual search in raw log data - using regular expressions to analyze log data (i.e., searching for messages that match a particular pattern) is tedious and often ineffective. Our argument is that analysts should not need to conduct a series of individual searches to find and in their mental map link relevant log lines at the time of analysis. What we propose, alternatively, is a knowledge extraction approach that includes regular expressions as part of the pipeline in order to facilitate entity recognition as one step of constructing a knowledge graph. Succinctly, we propose a shift from regex-based search to graph-pattern based querying (using regular expressions, amongst other techniques, in the extraction phase).
> ### Link to PROV
> We considered PROV-O as a candidate vocabulary, but found it difficult to apply it directly in this setting due to a few conceptual differences. The  modeling around agent, entity, and activity, for instance, did not quite fit as fine-granular provenance is not very common at the level of the extracted log data. We also considered using PROV for the modeling of the log sources as agents, but did not see important benefits of this less compact conceptualization. A partial alignment would certainly be possible, however.
> ### KG creation process
> The paper describes  - within the limited space available - the complete KG creation process, but we will update Fig. 1 with a new version (cf. [3]) that illustrates the process in more detail. Particularly,  the individual sub-steps in A2 (Template Enrichment) are included in this updated figure to make the overall process more transparent to the reader. Finally, we provide extensive documentation online (cf. [4])
> ### Configuration information and reuse
> We provide the complete source code, setup and documentation, all of which can be checked out [4] and reused by others - within and beyond the security example domain. The latter will require some moderate configuration effort; we explain the necessary steps in detail in the documentation [5].
> ### Related work
> We appreciate the recommendation and will move the related work section towards the end of the paper, and also remove the line pointing to future work from Table 1.
>
> [1] https://sepses.ifs.tuwien.ac.at/vocab/ref/cve/index-en.html#
> [2] https://github.com/Ebiquity/Unified-Cybersecurity-Ontology
> [3] https://raw.githubusercontent.com/sepses/slogert/master/slogert.jpg
> [4] https://github.com/sepses/slogert
> [5] https://github.com/sepses/slogert/blob/master/README.md

---

> > ### Comment · AnonReviewer4 · 2021-02-02
> > **Thanks for clarifying my questions**
> >
> > I think that the authors have done a good job answering my questions, and I think this is a promising approach. I increased my score to accept

---

### Official Review · AnonReviewer1 · 2021-01-19
**good approach, well written, poor valitation**

**Rating:** 1
**Confidence:** 4

**Review:**

This paper proposes SLOGERT, a system that automatically constructs a Knowledge Base from log messages. The resulting KGs enable analysts to navigate and query an integrated, enriched view of the events, thereby facilitating a novel approach for log analysis. Authors propose an extension of a prior vocabulary to express events (log:Event), and a set of predicates (e.g., log:hasParameter, log:parameter, log:hasSourceHost, log:hasLogSource) to handle relevant information about logs.

I enjoy seeing works like this, where we can use the tools that provide the Semantic Web ecosystem in another field of computer science. The paper is clear and well written. However, I expected more details about the extension of the vocabulary; for instance, is it available? Which are their relations with similar vocabularies, etc. Some articles advocate following good practices in the onology creation process (e.g., Baker et al. [1])


References
----------
[1] T. Baker, P. Vandenbussche, B. Vatant, Requirements for vocabulary preservation and governance, Library Hi Tech 31 (2013) 657–668.



**Anonymity:**

Yes, I would like my review to remain anonymous.

**Strong Points:**

- The authors are proposing a novel way to address the log analysis using semantic web mechanisms.
- Results show how they retrieve relevant information using SPARQL queries.

**Subreviewer:**

I submitted this review.

**Weak Points:**

I understand that there are no proper datasets to evaluate this approach because this is a novel perspective to deal with logs. Nonetheless, the paper's result shows how they retrieve concepts from AIT log dataset but do not provide any measurements about the quality of the proposed system.

---

> ### Author Rebuttal · Authors · 2021-01-29
>
> Thank you for the positive evaluation and the constructive feedback.
>
> ## Vocabularies
>
> ### Are the developed vocabularies available?
> The vocabularies are available and we do provide them in various serializations, as well as a Widoco documentation (cf. [1]  and [2]). Pointers to the vocabularies are provided in the paper (cf. footnotes 3 and 12, respectively).
>
> ### Relations with similar vocabularies
> To construct the core vocabulary used to represent the extracted log messages [2], we started from an existing minimal vocabulary for log representation [3] as well as CEE, a comprehensive taxonomy for the log domain [4] which we mapped into RDF and aligned with our base vocabulary. The vocabulary used to represent log templates extracted by SLOGERT was constructed specifically for the purpose.
>
> ### Ontology creation process
> The vocabulary design follows common Linked Data best practices as suggested by the reviewer, e.g., in terms of naming conventions, the consistent use of persistent, dereferenceable URIs for vocabulary terms, choice of representations, metadata and description, labels and definitions, reuse of existing well-known vocabularies etc. We will check and add a reference to the suggested requirements in the final version of the paper.
>
> ## Qualitative evaluation
> Regarding the remark that the evaluation does not cover qualitative aspects of the constructed graphs, we would like to emphasize that the completeness and quality of the constructed graphs hinges upon the configuration used. The published configuration (i.e., regex patterns, log extraction settings etc.) covers standard concepts we use for illustration purposes (logs relevant in a cybersecurity context); it will, however, have to be adapted for complete coverage and for different log domains (adaptation effort for the AIT dataset used in the evaluation was very low). Therefore, a key design goal in the development of SLOGERT was configurability and extensibility. Consequently, incomplete extraction templates and mappings can easily be adapted in an iterative process to improve the solution quality, e.g., by improving entity recognition on a specific log data source. In our evaluation use case, we added generic patterns to demonstrate the general usefulness [5], but did not strive to achieve completeness for a specific data set or log data source.
> Even though a complete qualitative evaluation is beyond the scope of the paper, we can report on the number of correctly detected templates and the number of correctly extracted parameter types in our use case.
>
> [1] https://w3id.org/sepses/ns/log#
> [2] https://w3id.org/sepses/ns/logex#
> [3] https://w3id.org/sepses/vocab/log/core
> [4] https://cee.mitre.org/language/1.0-beta1/core-profile.html
> [5] https://github.com/sepses/slogert/blob/master/src/main/resources/config.yaml

---

### Decision · Program_Chairs · 2021-02-23

**Decision:**

Accept

**Comment:**

All reviewers agree that this paper addresses an important problem that is not often tackled by the Semantic Web community. Some concerns have been expressed about the maturity of the project, whose reusability may be limited in its current state. However, given the potential impact of the approach, the recommendation is to accept it ­– provided that the authors address the suggestions of the reviewers, as they have started to do in the rebuttal.